# Exosomal miR-205-5p Improves Endometrial Receptivity by Upregulating E-Cadherin Expression through ZEB1 Inhibition

**DOI:** 10.3390/ijms242015149

**Published:** 2023-10-13

**Authors:** Seong-Lan Yu, Da-Un Jeong, Eui-Jeong Noh, Hye Jin Jeon, Dong Chul Lee, Minho Kang, Tae-Hyun Kim, Sung Ki Lee, Ae Ra Han, Jaeku Kang, Seok-Rae Park

**Affiliations:** 1Priority Research Center, Myunggok Medical Research Institute, College of Medicine, Konyang University, Daejeon 35365, Republic of Korea; yusl73@konyang.ac.kr (S.-L.Y.); ekdns493@naver.com (D.-U.J.); 22kyujhj@gmail.com (H.J.J.); th2580@kyuh.ac.kr (T.-H.K.); sklee@kyuh.ac.kr (S.K.L.); 2Department of Obstetrics and Gynecology, College of Medicine, Konyang University, Daejeon 35365, Republic of Korea; no205090@naver.com; 3Personalized Genomic Medicine Research Center, Korea Research Institute of Bioscience and Biotechnology, Daejeon 34141, Republic of Korea; dclee@kribb.re.kr (D.C.L.); mhkang@kribb.re.kr (M.K.); 4Department of Functional Genomics, KRIBB School of Bioscience, University of Science and Technology, Daejeon 34141, Republic of Korea; 5Department of Obstetrics and Gynecology, Konyang University Hospital, Daejeon 35365, Republic of Korea; 6I-Dream Clinic, Department of Obstetrics and Gynecology, Mizmedi Hospital, Seoul 07639, Republic of Korea; sonagy1@naver.com; 7Daegu CHA Fertility Center, CHA University, Daegu 42469, Republic of Korea; 8Department of Pharmacology, College of Medicine, Konyang University, Daejeon 35365, Republic of Korea; 9Department of Microbiology, College of Medicine, Konyang University, Daejeon 35365, Republic of Korea

**Keywords:** endometrial receptivity, exosome, exosomal miRNA, miR-205-5p/ZEB1/E-cadherin axis, embryo implantation

## Abstract

Endometrial receptivity is a complex process that prepares the uterine endometrium for embryo implantation; insufficient endometrial receptivity is one of the causes of implantation failure. Here, we analyzed the microRNA expression profiles of exosomes derived from both receptive (RL95-2) and non-receptive (AN3-CA) endometrial epithelial cell (EEC) lines to identify exosomal miRNAs closely linked to endometrial receptivity. Among the 466 differentially expressed miRNAs, miR-205-5p was the most highly expressed in exosomes secreted from receptive RL95-2 cells. miR-205-5p, enriched at the adhesive junction, was closely related to endometrial receptivity. ZEB1, a transcriptional repressor of E-cadherin associated with endometrial receptivity, was identified as a direct target of miR-205-5p. miR-205-5p expression was significantly lower in the endometrial tissues of infertile women than in that of non-infertile women. In vivo, miR-205-5p expression was upregulated in the post-ovulatory phase, and its inhibitor reduced embryo implantation. Furthermore, administration of genetically modified exosomes overexpressing miR-205-5p mimics upregulated E-cadherin expression by targeting ZEB1 and improved spheroid attachment of non-receptive AN3-CA cells. These results suggest that the miR-205-5p/ZEB1/E-cadherin axis plays an important role in regulating endometrial receptivity. Thus, the use of exosomes harboring miR-205-5p mimics can be considered a potential therapeutic approach for improving embryo implantation.

## 1. Introduction

Establishment of pregnancy requires the successful implantation of a competent blastocyst into a receptive endometrium. The human endometrium comprises the basalis and functionalis, which are the innermost layers of the uterus. The functionalis undergoes dynamic morphological changes such as regeneration, differentiation, and shedding during the menstrual cycle. The endometrium needs to be receptive to embryo implantation during the mid-secretory phase, known as the window of implantation (WOI), which is regulated by the sequential action of steroid hormones such as estrogen and progesterone [1,2,3]. Approximately one-third of implantation failures are caused by insufficient endometrial receptivity [4]. Despite the development of assisted reproductive technology (ART), 60–70% of women fail to achieve pregnancy after the transfer of good-quality embryos [5]. Therefore, several studies have investigated the molecular mechanisms involved in transforming the receptive endometrium as a prerequisite for successful embryo implantation.

MicroRNAs (miRNAs), composed of approximately 19–22 nucleotides, are small single-stranded non-coding RNAs that play important roles in RNA silencing and post-transcriptional repression of target mRNAs via complementary base paring of the 3′ untranslated region (3′ UTR) [6]. Studies have reported on miRNAs associated with endometrial receptivity and embryo implantation [7,8]. miRNAs exist widely in cells and tissues; extracellular miRNAs have also been identified in various biological fluids, including peripheral blood, serum, and milk, and cell culture supernatants, where they are packaged into extracellular vehicles (EVs) [9,10].

Exosomes, a type of lipid membrane-bound EVs, are smaller (approximately 30–150 nm in diameter) than most other EVs and are released by different cells. They contain various molecules, such as proteins, lipids, and nucleic acids, and play critical roles in intercellular communication by transporting these molecules from donor cells to recipient cells [11]. Ng et al. isolated exosomes from uterine fluid and suggested the possibility of transferring molecules associated with implantation to the embryo through exosomes released from the endometrial epithelium [12]. The expression of proteins in endometrial exosomes is regulated by hormones that modulate the menstrual cycle; notably, endometrial exosomes of the secretory phase are enriched with proteins known to be involved in embryo implantation [13]. During the peri-implantation period, intrauterine exosomes are essential for successful embryo implantation [14]. Recently, miRNAs from endometrial cell-derived EVs have been found to be differentially expressed between the pre-receptive and receptive stages; a few miRNAs have also been reported as being relevant to embryo implantation [15,16].

In this study, we performed miRNA expression profiling of exosomes derived from both receptive (RL95-2) and non-receptive (AN3-CA) endometrial epithelial cell (EEC) lines and identified miR-205-5p to be closely associated with endometrial receptivity. Furthermore, we investigated the molecular mechanisms underlying the positive effects of miR-205-5p on endometrial receptivity both in vitro and in vivo.

## 2. Results

### 2.1. Characterization of Exosomes Derived from EECs Reveals Differences in Endometrial Receptivity

RL95-2 cells have been widely used as an in vitro model for human receptive uterine epithelium because of their high adhesiveness to trophoblast-derived cells or spheroids, whereas AN3-CA cells have been considered to represent non-receptive uterine epithelium because of their low adherence [17,18,19]. Our previous study also confirmed differences in endometrial receptivity rates in an in vitro experimental model [20]. To elucidate the mechanisms underlying these differences between the cell lines, we focused on identifying exosomal miRNAs that affect endometrial receptivity. The morphology of the isolated exosomes was confirmed using cryo-electron microscopy (cryo-EM). As shown in Figure 1a, exosomes derived from the two EECs were round in shape and similar to each other. Exosomes secreted from the two cell lines that exhibited differences in receptivity were isolated and confirmed using Western blotting with exosome markers. Although the expression level of CD63 differed between the cell lines, the level of CD9, an exosome surface marker, was similar (Figure 1b). These results indicate that cell-derived exosomes of high quality and purity were successfully isolated. Particle size distribution based on nanoparticle tracking analysis (NTA) revealed that the exosomes derived from the two EECs were physically homogenous particles with a peak at approximately 90 nm and an average diameter of approximately 172 nm (Figure 1c). The isolated exosomes derived from RL95-2 cells were fluorescently labeled and incubated with AN3-CA recipient cells. The labeled exosomes were found to be localized to the cytoplasm of recipient cells, indicating that exosomes derived from RL95-2 cells were efficiently incorporated into the recipient cells (Figure 1d). Next, we investigated the effect of exosomes derived from receptive RL95-2 cells on spheroid adhesion to non-receptive AN3-CA cells. RL95-2 cell-derived exosomes showed approximately 10% improvement in spheroid attachment compared to untreated control cells (Figure 1d). These findings indicate that improved spheroid attachment may be related to biological molecules within exosomes secreted from receptive RL95-2 cells.

### 2.2. Identification of Differentially Expressed miRNAs in EEC-Derived Exosomes

To identify the exosome-derived biological molecules involved in spheroid attachment, we performed miRNA expression profiling of the exosomes derived from the two EECs that exhibited differences in endometrial receptivity. Next-generation sequencing (NGS) revealed 466 significantly differentially expressed miRNAs based on a threshold of [log FC] > 1.5 (*p* < 0.05). A volcano plot was used to display differences in the expression levels of differentially expressed miRNAs in the EEC-derived exosomes. Of these 466 miRNAs, 267 were upregulated and 198 were downregulated in exosomes derived from RL95-2 cells compared with those derived from AN3-CA cells (Figure 2a). The top 20 upregulated and downregulated miRNAs are displayed in a hierarchical clustering heatmap (Figure 2b and Table 1). Among the differentially expressed miRNAs, miR-205-5p was the most highly expressed in exosomes secreted from RL95-2 cells, whereas miR-196a-5p was the least expressed. Four differentially expressed miRNAs were selected, and their expression patterns were confirmed using quantitative reverse transcription polymerase chain reaction (qRT-PCR) analysis. As shown in Figure 2c, the expression of miR-205-5p and miR-200c-3p was significantly upregulated in both RL95-2 cells and RL95-2 cell-derived exosomes. In contrast, miR-196a-5p and miR-615-3p were significantly downregulated in both RL95-2 cells and RL95-2 cell-derived exosomes (Figure 2d). The results of the qRT-PCR analysis were consistent with the expression profiling results obtained using NGS. Next, we analyzed the Kyoto Encyclopedia of Genes and Genomes (KEGG) pathways of the target genes to understand the potential roles of the four selected miRNAs (Figure 2e). Significant pathways with *p* < 0.05 were determined using a Fisher’s exact test and DIANA mirPath v.3 (TarBase v7.0 method). An adherens junction, closely related to endometrial receptivity, was categorized as a potential function of miR-205-5p and miR-200c-3p. These results suggest that miR-205-5p, the most highly expressed miRNA in exosomes secreted from RL95-2 cells, which represent a highly receptive endometrial epithelium, may be involved in the adhesion process of endometrial cells.

### 2.3. Identification of miR-205-5p Target Genes Related to the Adhesion of Endometrial Cells

To investigate the molecular mechanisms underlying the promotion of exosome-induced spheroid attachment, the potential target genes of miR-205-5p were analyzed using bioinformatics. Based on analyses of the miRDB, TargetScan, and MicroT-CDS databases, 249 genes were identified as potential targets (Figure 3a). Among these putative targets, genes involved in regulating adhesive junctions associated with endometrial receptivity were determined. We chose ZEB1 for further analysis because it has been reported to be a transcriptional repressor of E-cadherin, an important protein in the adherent junction process [21]. E-cadherin also plays a critical role in the initial attachment of embryos [22]. To confirm whether ZEB1 is a potential target gene of miR-205-5p, we searched for the miR-205-5p binding site in the ZEB1 mRNA sequence. The binding site for miR-205-5p was predicted to be between nucleotide (nt) 4470 and 4477 in the 3′ UTR of the ZEB1 mRNA (NM_001128128; Figure 3b). A dual luciferase reporter assay was then performed to verify the direct interaction between miR-205-5p and the 3′ UTR of ZEB1. Overexpression of miR-205-5p mimics significantly reduced the luciferase activity of the reporter construct carrying ZEB1-3′ UTR-WT, containing the potential miR-205-5p binding site. However, miR-205-5p mimics did not affect the activity of the ZEB1-3′ UTR-MUT reporter construct in which the miR-205-5p binding site in the ZEB1 3′ UTR was mutated (Figure 3c). Subsequently, qRT-PCR and Western blotting were performed to measure ZEB1 mRNA and protein expression in AN3-CA cells transfected with the miR-205-5p mimics or miRNA negative control (miR-NC). The transfection efficiency of the miR-205-5p mimics was first confirmed (Figure 3d). ZEB1 mRNA expression was significantly decreased following transfection with miR-205-5p mimics compared with that in the miR-NC-transfected cells (Figure 3e). The ZEB1 protein level also decreased in miR-205-5p mimic-transfected AN3-CA cells (Figure 3f), indicating that miR-205-5p inhibited ZEB1 expression by targeting the 3′ UTR. To further confirm the relationship between miR-205-5p and ZEB1, we analyzed the expression of CDH1, a target gene repressed by ZEB1. CDH1 mRNA expression was clearly upregulated by the overexpression of miR-205-5p mimics (Figure 3g). The alteration in E-cadherin protein expression was consistent with the RNA expression patterns (Figure 3h). These results suggest that miR-205-5p positively regulates E-cadherin expression by targeting ZEB1.

### 2.4. Role of miR-205-5p in Embryo Implantation

To determine the relationship between miR-205-5p and endometrial receptivity, we checked its expression pattern in endometrial samples from non-infertile and infertile women. The expression of miR-205-5p was significantly lower in infertile women than in non-infertile women (Figure 4a). In contrast, ZEB1 mRNA expression was relatively high in infertile tissues (Figure 4b). The negative correlation between miR-205-5p and ZEB1 was consistent with the findings from the cell line models. These results indicate that the miR-205-5p/ZEB1 axis may be closely associated with endometrial receptivity, a crucial event in the establishment of pregnancy. As a receptive endometrium is required for successful embryo implantation, we used mouse models at different phases of the reproductive cycle to explore the function of miR-205-5p during embryo implantation. Mouse uteri were collected before and after ovulation to validate the expression pattern of miR-205-5p. The expression of miR-205-5p was significantly upregulated in the post-ovulatory phase compared with that in the pre-ovulatory phase (Figure 4c), suggesting that miR-205-5p may be involved in embryo implantation. To demonstrate this possibility, mice were randomly divided into three groups. Next, PBS, an miR-NC inhibitor, or an miR-205-5p inhibitor was injected into the uterine horn of a 2.5 days post coitum (dpc) pregnant uterus. At 12.5 dpc, the mice were sacrificed, and the number of implanted embryos was recorded. The number of implanted embryos was lower in the miR-205-5p inhibitor-treated group than in the miR-NC inhibitor and PBS groups (Figure 4d). There was no difference in the number of implanted embryos between the miR-NC inhibitor and PBS groups, confirming that the injection of inhibitors did not affect embryo implantation. Therefore, miR-205-5p may play an important role in embryo implantation by regulating endometrial receptivity.

### 2.5. Role of Modified Exosomes Overexpressing miR-205-5p Mimics in Endometrial Receptivity

To further verify the function of miR-205-5p in improving endometrial receptivity, we generated genetically modified exosomes overexpressing miR-205-5p mimics (miR-205-5pOE). miR-205-5p mimics were transfected into exosomes derived from non-receptive AN3-CA cells expressing low levels of miR-205-5p. Exosomes modified with miR-205-5p mimics were then administered to AN3-CA cells to confirm their functional significance in endometrial receptivity. The miR-205-5p level significantly increased in AN3-CA cells treated with modified miR-205-5pOE exosomes (Figure 5a). ZEB1 mRNA, a potential target gene of miR-205-5p, was reduced by the overexpression of miR-205-5p mimics (Figure 5b). The protein level of ZEB1 also decreased in the miR-205-5pOE exosome-treated cells (Figure 5c). Furthermore, the exosome-mediated transfer of miR-205-5p resulted in a significant increase in E-cadherin mRNA and protein expression, as shown in Figure 5d,e. We also visualized increased E-cadherin expression in miR-205-5pOE exosome-treated AN3-CA cells using immunofluorescence analysis (Figure 5f). To determine whether the exosome-mediated transfer of miR-205-5p mimics improved endometrial receptivity, we performed an in vitro spheroid attachment assay. The spheroid attachment of AN3-CA cells was improved by modified exosomes overexpressing the miR-205-5p mimics (Figure 5g). These results strongly suggest that the miR-205-5p/ZEB1/E-cadherin axis is involved in regulating endometrial receptivity. Furthermore, the exosome-mediated transfer of miR-205-5p can improve the endometrial receptivity of non-receptive EECs by regulating the ZEB1/E-cadherin signaling cascade. The results of this study are summarized in Figure 6.

## 3. Discussion

Successful embryo implantation is an essential step for establishing pregnancy and involves interactions between the endometrium and embryo. The embryo is only accepted by the endometrium in a receptive state, known as the “WOI”, which changes the pre-receptive state into a receptive endometrium through the alteration of several molecular mediators [23,24]. Studies have reported differences between pre-receptive and receptive endometrial transcriptomes to investigate the molecular mechanisms associated with endometrial receptivity [25,26,27,28,29,30,31,32,33]. In this study, we explored the differentially expressed miRNAs in exosomes derived from receptive (RL95-2) and non-receptive (AN3-CA) EECs to understand the mechanism involved in improved endometrial receptivity during implantation. Among 466 differentially expressed miRNAs, miR-205-5p was selected using NGS and bioinformatic methods. Recently, Yang et al. (2023) reported miR-205-5p expression in uterus fluid [34]. The cellular and molecular mechanisms underlying the association between miR-205-5p and endometrial receptivity were investigated using in vitro and in vivo models.

When analyzing the KEGG pathways enriched by the target genes for further study, two selected miRNAs were found to be associated with the dynamics of the endometrium, such as the adherens junction. The adherens junctions are direct cell–cell contacts that maintain cell and tissue polarity mediated by cadherin and catenin family proteins. Among cadherins, E-cadherin is the most abundant in the adherens junctions of epithelial cells and interacts with cytoplasmic catenin proteins [35]. E-cadherin has been identified in both trophoblasts and the endometrial epithelium, and its expression is substantially upregulated in the secretory phase of the uterine endometrium [36,37]. Moreover, E-cadherin has been designated as an endometrial receptivity marker because of its critical role in the initial attachment process during embryo implantation [19,22,38]. Of these two miRNAs, we selected miR-205-5p, which is highly expressed in exosomes secreted from RL95-2 cells, which represent a highly receptive endometrial cell line (Figure 2). Additionally, miR-205-5p expression was found to be significantly upregulated in the post-ovulatory phase in mice. However, the expression of miR-205-5p was significantly lower in the endometrial tissues of infertile women than in that of non-infertile women (Figure 4). This finding suggests that the expression pattern of miR-205-5p is closely related to endometrial receptivity. In previous studies, miR-205-5p has been found to be negatively associated with epithelial–mesenchymal transition (EMT) in several cancers, including breast cancer, liver cancer, glioblastoma, thyroid carcinoma, and colon cancer [39,40,41,42,43,44,45]. EMT is a process that changes the morphology of cells from epithelial cells, which maintain cell polarity and cell–cell adhesion through tight junctions, adherens junctions, and gap junctions, to mesenchymal cells. In particular, loss of E-cadherin, a well-known epithelial cell marker, is considered an essential event in EMT [46]. Several studies have reported that miR-205-5p upregulates E-cadherin expression by inhibiting ZEB1 expression [41,42,43]. Moreover, suppressing ZEB1 expression induces E-cadherin expression in endometrial epithelial cells such as Ishikawa and HEC-1B cells [47]. We also found that miR-205-5p downregulated ZEB1 expression by directly binding to its 3′ UTR and induced E-cadherin expression through overexpression of miR-205-5p in AN3-CA cells (Figure 3). Mice are widely used animal models to study human uterine receptivity for embryo implantation. In our study, miR-205-5p expression was significantly upregulated in the post-ovulatory phase compared with that in the pre-ovulatory phase. As expected, injection of an miR-205-5p inhibitor into the uterus during the WOI led to a reduction in the embryo implantation rate (Figure 4). These results indicate that miR-205-5p may play an important role in embryo implantation by regulating endometrial receptivity. Therefore, exosomal miR-205-5p may serve as a novel biomarker for endometrial receptivity.

We identified studies on endometrial regulation involving selected miRNAs, including miR-200c, miR-196a, and miR-615-3p. miR-200c, the expression of which is upregulated in receptive cell-derived exosomes, is reportedly relevant to endometrial development during early pregnancy [48]. Moreover, miR-200c reduces ZEB1 expression and induces E-cadherin expression in human endometrial stromal cells. It also reduces the growth of ectopic endometrial cysts in a rat model [49]. However, Zheng et al. reported opposite results, suggesting that miR-200c plays a role in the inhibition of endometrial receptivity and embryo implantation [50]. miR-196a, the expression of which is downregulated in receptive cell-derived exosomes, has been reported to be upregulated in the eutopic endometrium of patients with endometriosis compared to the control endometrium. It represses the decidualization of endometrial stromal cells [51]. Furthermore, Zhu et al. recently suggested that miR-196a-5p expression is upregulated by estrogen and that it targets FOXO1 in endometrial epithelial cells, such as RL95-2, HEC-1, and ECC-1 cells [52]. However, the role of miR-615-3p in endometrial epithelial cells remains to be explored. These findings suggest the importance of studying multiple miRNAs in the endometrium to understand their roles in endometrial receptivity and embryo implantation.

Recently, Zhang et al. reported that treatment with exosomes derived from human umbilical cord mesenchymal stem cells significantly increased the endometrial thickness, expression of molecular markers of endometrial receptivity, and pregnancy rates in a rat model of thin endometrium [53]. Similarly, exosomes derived from bone marrow stem cells (BMSCs) and genetically modified BMSCs promote the regeneration of endometrial tissues and improve endometrial receptivity in a rat model [54]. Based on these findings, miR-205-5p mimics were transfected into exosomes derived from non-receptive AN3-CA cells to confirm their functional importance in endometrial receptivity. After cellular uptake of exosomes modified with miR-205-5p mimics, the resulting overexpression of miR-205-5p mimics reduced ZEB1 mRNA expression and increased E-cadherin expression in non-receptive AN3-CA cells. Furthermore, spheroid attachment of AN3-CA cells was clearly improved by exosomes overexpressing miR-205-5p mimics (Figure 5). Thus, our results suggest that exosome-mediated transfer of miR-205-5p mimics can improve the endometrial receptivity of non-receptive EECs by regulating the ZEB1/E-cadherin signaling cascade. Therefore, similar to the findings of previous studies, exosomes containing miR-205-5p can be proposed as an exosome-based therapeutic to improve embryo implantation (Figure 6).

In conclusion, exosomes derived from receptive and non-receptive EECs contain different miRNAs that may be associated with endometrial receptivity. Specifically, miR-205-5p is highly expressed in exosomes secreted from receptive RL95-2 cells and upregulates E-cadherin expression by targeting ZEB1. Upon treatment with exosomes overexpressing miR-205-5p mimics, improved endometrial receptivity was observed in non-receptive AN3-CA cells. Therefore, the miR-205-5p/ZEB1/E-cadherin axis represents a potential therapeutic target to improve endometrial receptivity and the success of embryo implantation. Further studies are needed to explore the clinical applications of exosome-based therapies targeting this axis in the context of implantation failures.

## 4. Materials and Methods

### 4.1. Collection of Human Endometrial Tissue

Human endometrial samples of the secretory phase were obtained from infertile women at MizMedi Hospital and non-infertile women at Konyang University Hospital, respectively. Infertility is defined as the inability of a couple to conceive naturally even after 1 year of regular unprotected sexual intercourse [55]. Infertile patients were selected according to this definition, and their diagnosis history was as follows. Two of six selected infertile patients experienced two or more repeated miscarriages. Three patients had at least three high-quality embryos implanted into the uterus but did not become pregnant. Therefore, they were diagnosed with recurrent implantation failure. In addition, one patient was diagnosed with recurrent miscarriages and recurrent implantation failures. Non-infertile patients were recruited for a clinical trial at Konyang University Hospital, and they provided written informed consent for the collection of endometrial tissues. The menstrual stages of the samples were determined by experienced gynecological pathologists using the Noyes criteria [56]. This study was approved by the bioethics committees of Konyang University Hospital (IRB file No. 2018-11-007) and MizMedi Hospital (IRB file No. 2018-3). The characteristics of the participants’ endometria are shown in Table 2.

### 4.2. Cell Culture

Human EECs (AN3-CA and RL95-2) were obtained from the American Type Culture Collection (Manassas, VA, USA), and human choriocarcinoma JAr cells were obtained from the Korea Cell Line Bank (Seoul, Korea). AN3-CA, RL95-2, and JAr cells were cultured in MEM, DMEM/F-12, and RPMI-1640 (Hyclone, Logan, UT, USA), respectively. All media were supplemented with 10% fetal bovine serum (FBS; Gibco, Waltham, MA, USA) and 1% penicillin–streptomycin (Hyclone). These cells were maintained at 37 °C in a humidified atmosphere containing 5% CO_2_.

### 4.3. Exosome Isolation, Uptake, and Transfection

Exosomes were isolated from cell culture media using ExoQuick-TC according to the manufacturer’s instructions (System Biosciences, Palo Alto, CA, USA) and stored at −80 °C until use.

To examine the uptake of exosomes by cells, exosomes were first labeled using the ExoGlow™-protein EV labeling kit (System Bioscience). The cells were treated with the labeled exosomes for 24 h, fixed using paraformaldehyde (Sigma, St. Louis, MO, USA), and imaged using a confocal laser scanning microscope (Carl Zeiss, Oberkochen, Germany).

Transfection of the miR-NC or miR-205-5p mimics into exosomes was performed using Exo-Fect™ according to the manufacturer’s protocol (System Biosciences).

### 4.4. Cryo-EM and NTA

Cryo-EM was performed to directly visualize the exosomes. Briefly, 3 μL of the exosome suspension was applied to Quantifoil holey carbon EM grids (R1.2/1.3, 200 mesh; EMS). The EM grids were then glow-discharged for 60 s at 15 mA before sample application. Furthermore, the grids were blotted with Vitrobot Mark IV (FEI, Hillsboro, OT, USA) for 3 s with 100% relative humidity at 4 °C. The samples were then imaged using Glacios (FEI) at an acceleration voltage of 200 kV with a Falcon IV direct electron detector (FEI). Images were captured at 120,000× magnification and −3.0-μm defocus.

The size distribution of exosomes was determined via NTA using a Nanosight NS300 instrument with NTA2.3 analytical software (Malvern Panalytical Ltd., Malvern, UK).

### 4.5. NGS Library Generation and Sequencing

NGS libraries were generated using the TailorMix Micro RNA Sample Preparation version 2 protocol (SeqMatic LLC, Fremont, CA, USA). Briefly, the 3′-adapter was ligated to the RNA sample and the excess was subsequently removed. The 5′-adapter was then ligated to 3′-adapter-ligated samples, followed by first-strand cDNA synthesis. The cDNA library was amplified and barcoded using enrichment PCR. The final RNA library was size selected using an 8% TBE polyacrylamide gel. Sequencing was performed on an Illumina NextSeq 500 platform at read lengths of 1 × 75 bp single-end and SR50.

### 4.6. Immunoblot Analysis

The cells were lysed on ice in RIPA buffer (Jubiotech, Daejeon, Korea) containing protease and phosphatase inhibitors (Roche, Basel, Switzerland). The protein concentration of the lysates was measured using the bicinchoninic acid (BCA) assay (Thermo Fisher Scientific, Waltham, MA, USA). Proteins were resolved using sodium dodecyl sulfate-polyacrylamide gel electrophoresis (SDS-PAGE) and transferred onto polyvinylidene difluoride membranes (PVDF; Millipore, Burlington, MA, USA). The blots were blocked with 5% skim milk (Difco, Detroit, MI, USA) and probed overnight with primary antibodies against CD9, CD63 (Abcam, Cambridge, UK), E-cadherin, ZEB1, and GAPDH (Cell Signaling Technology, Danvers, MA, USA). On the following day, the membranes were incubated with horseradish peroxidase-conjugated secondary antibodies (Millipore, Burlington, MA, USA). The immunoreactive proteins were detected using an Enhanced Chemiluminescence Kit (Thermo Fisher Scientific).

### 4.7. Total and Exosomal RNA Isolation and qRT-PCR Analysis

Total RNA was extracted from cells and endometrial tissue biopsies using TRIzol^®^ reagent (Ambion, Thermo Fisher Scientific, Waltham, MA, USA) according to the manufacturer’s instructions. To determine mRNA expression, total RNA was reverse transcribed to cDNA using Moloney murine leukemia virus reverse transcriptase (Promega, Madison, WI, USA). Quantitative real-time PCR was performed in triplicate using iQ SYBR Green Supermix and a CFX Connect Real-Time PCR Detection System (Bio-Rad Laboratories, Hercules, CA, USA). The primer sets used are listed in Table 3. The 2^−ΔΔCt^ method was used to calculate mRNA expression levels based on the reference gene.

Exosomal RNA was isolated using an ExoQuick^®^ exosome RNA purification kit (System Biosciences) according to the manufacturer’s instructions. To investigate the relative miRNA expression, cDNA was synthesized using the TaqMan miRNA Reverse Transcription Kit and TaqMan miRNA RT primers (Thermo Fisher Scientific) according to the manufacturer’s instructions. Quantitative real-time PCR was performed using TaqMan Master Mix II and TaqMan miRNA assay primers (Thermo Fisher Scientific). The 2^−ΔΔCt^ method was used to calculate miRNA expression levels based on the reference miRNA.

### 4.8. In Vitro Assay for JAr Spheroid Implantation

JAr cells were seeded in a V-bottom microplate (Greiner Bio-one, Kremsmünster, Austria) to form JAr cell spheroids and incubated in culture media for 24 h in a humidified incubator containing 5% CO_2_. The endometrial epithelial cells were cultured in 24-well plates until they reached 90% confluence and treated with exosomes for 48 h. The spheroids were then harvested and co-cultured for 6 h with monolayer endometrial epithelial cells and miR-205-5p- or miR-NC-overexpressing exosomes derived from AN3-CA cells. To count the attached spheroids, the plate was inverted, centrifuged at 10× *g* for 10 min, and the attached spheroids were then counted under a microscope (Olympus, Center Valley, PA, USA).

### 4.9. Plasmid Construction and Luciferase Reporter Assay

We identified the binding site of miR-205-5p on the ZEB1 3′ UTR using the miRDB (http://mirdb.org, accessed on 6 March 2023), Targetscan (https://www.targetscan.org/vert_80/, accessed on 6 March 2023), and microT-CDS (https://dianalab.e-ce.uth.gr/html/dianauniverse/index.php?r=microT_CDS, accessed on 6 March 2023) databases and amplified the ZEB1 3′ UTR with a forward primer containing an XhoI restriction site (5′-AAACTCGAGGTGTGCCTGAACCTCAGACC-3′) and a reverse primer containing a NotI restriction site (5′-AAAGCGGCCGCTCTCAATGCAGGAGAACCAA-3′). The ZEB1 3′ UTR was then cloned into a dual-luciferase psiCHECK2 vector (Promega, USA). Mutagenesis of the miR-205-5p binding site was performed using the KOD Plus Mutagenesis Kit (Toyobo, Osaka, Japan). The wild-type and mutant clones were verified via sequencing. To identify whether miR-205-5p modulates the ZEB1 3’ UTR, the cells were seeded in a plate and co-transfected with the cloned vector and miR-205-5p mimics or miR-NC using Lipofectamine 3000 (Thermo Fisher Scientific). Luciferase activity was then measured using a Dual-Luciferase Reporter Assay System (Promega). The luminescent activity of firefly luciferase was used as an internal control for normalizing transfection efficiency.

### 4.10. Animal Experiments

To examine whether miR-205-5p affects embryo implantation, 6–8-week-old C57BL/6 mice were purchased from DBL (DBL Co., Ltd., Chungbuk, Korea). Uteri at different estrus cycles were collected to investigate miR-205-5p expression. Pregnant mice were randomly divided into the following three groups: PBS, miR-NC inhibitor, and miR-205-5p inhibitor. In the PBS, miR-NC inhibitor, and miR-205-5p inhibitor groups, the right uterine horns were injected with PBS, mmu-miR-NC inhibitor, and mmu-miR-205-5p, respectively, at 2.5 dpc. At 12.5 dpc, the mice were sacrificed and the number of implantation sites was recorded. All animal experiments were approved by the institutional animal care and use committee of Konyang University (approval no. KY-IACUC-P22-29-E-01).

### 4.11. Statistical Analyses

All experiments were independently performed thrice, and the data are presented as mean ± standard error of the mean (SEM). The results were analyzed using a Student’s *t*-test or Mann–Whitney test. Statistical significance was considered at *p* < 0.05 and *p* < 0.01.

## Figures and Tables

**Figure 1 ijms-24-15149-f001:**
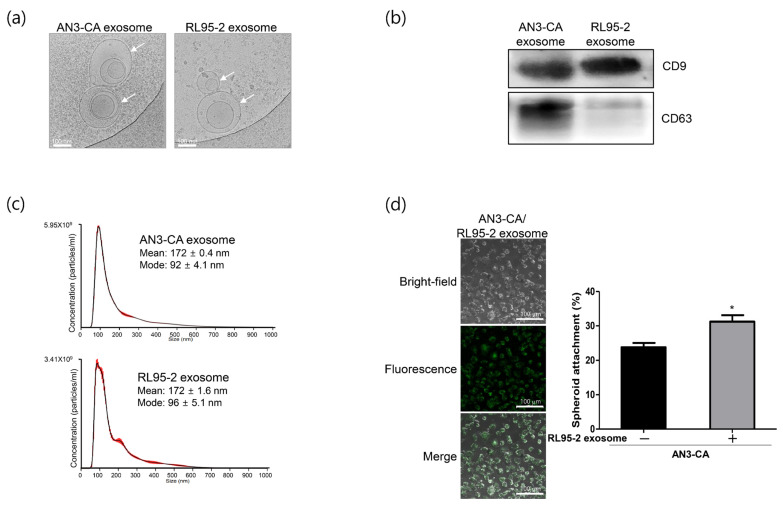
Isolation and characterization of exosomes derived from AN3-CA and RL95-2 endometrial epithelial cells (EECs). (**a**) Cryo-electron microscopy image of exosomes derived from AN3-CA and RL95-2 EECs. Exosome images are shown at 120,000× magnification. Scale bars indicate 100 nm. White arrows indicate exosomes. (**b**) Western blot analysis of the expression of the exosome-specific markers CD9 and CD63 in isolated exosomes. (**c**) Size and concentration analyses of isolated exosomes based on nanoparticle tracking analysis (NTA). (**d**) Confocal images showing cellular uptake of fluorescence-labeled exosomes derived from RL95-2 cells in AN3-CA cells. Spheroid attachment rate of AN3-CA cells treated with RL95-2 cell-derived exosomes. Images of exosome uptake are shown at 400× magnification. Scale bars indicate 100 µm. The data shown represent mean ± SEM from three independent experiments (*n* = 3). * *p* < 0.05.

**Figure 2 ijms-24-15149-f002:**
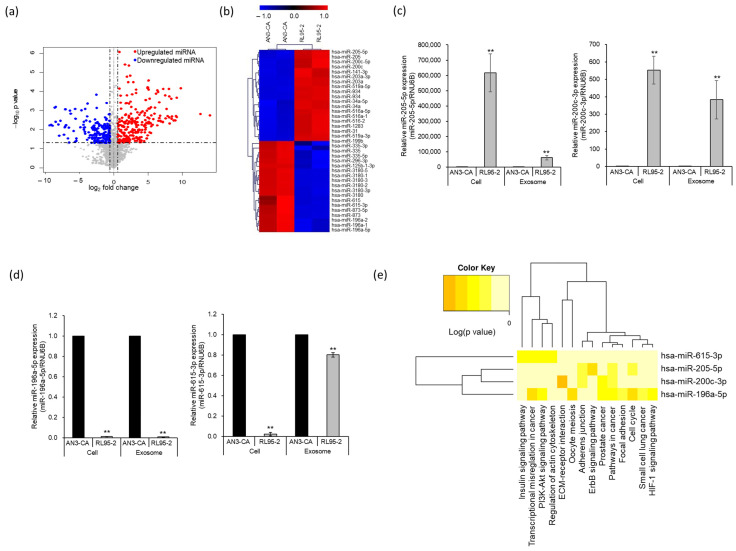
Identification of differentially expressed miRNAs between AN3-CA and RL95-2 endometrial epithelial cell-derived exosomes. (**a**) Volcano plot displaying differentially expressed miRNAs. Red dots represent significantly upregulated miRNAs; blue dots represent significantly downregulated miRNAs. (**b**) Hierarchical clustering heatmap of the top 20 upregulated and downregulated miRNAs. Red and blue colors indicate high and low expression of each miRNA, respectively. (**c**,**d**) Gene expression analysis of four selected miRNAs using quantitative RT-PCR (qRT-PCR) in exosomes and cell lines. The data shown represent mean ± SEM from three independent experiments (*n* = 3). ** *p* < 0.01. (**e**) KEGG pathway analysis of miRNAs validated using qRT-PCR. Significant pathways generated by DIANA-miRPath v3.0 software using the Tarbase database are represented on the x-axis and miRNAs are presented on the y-axis of the heat map.

**Figure 3 ijms-24-15149-f003:**
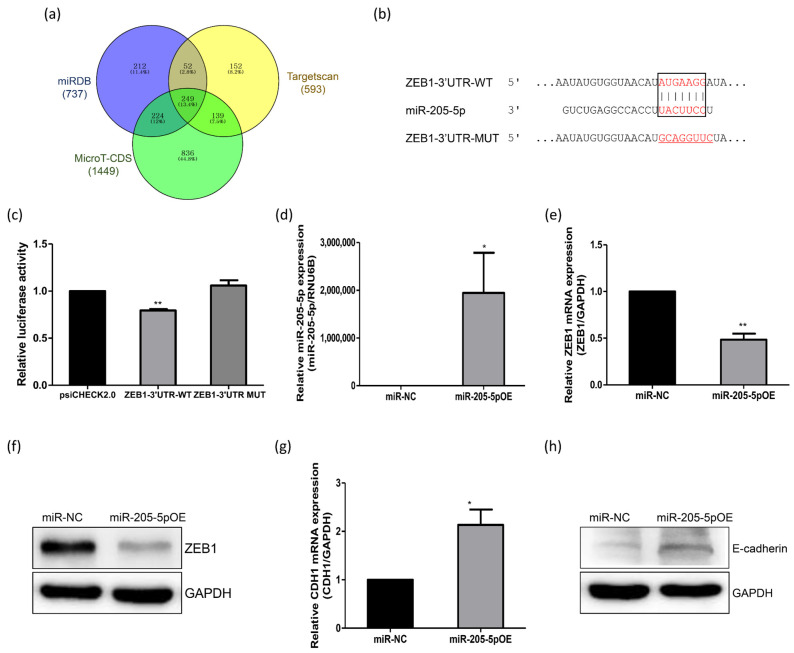
miR-205-5p directly targets the 3′ untranslated region (UTR) of ZEB1. (**a**) Venn diagram comparing target gene predictions using the miRDB, Targetscan, and MicroT-CDS databases for miR-205-5p. (**b**) Schematic representation of the predicted target sequence of miR-205-5p and the mutant between nt 4470 and 4477 in the 3′ UTR of *ZEB1*. (**c**) Relative luciferase activity of the ZEB1 3′ UTR and mutant reporter by overexpression of miR-205-5p mimics. (**d**) Expression analysis of miR-205-5p in AN3-CA cells treated with miR-205-5p mimics. (**e**) Quantitative RT-PCR analysis to detect *ZEB1* mRNA in AN3-CA cells treated with miR-205-5p mimics. (**f**) Western blot analysis to detect ZEB1 protein in AN3-CA cells treated with miR-205-5p mimics. (**g**) Quantitative RT-PCR analysis to detect CDH1 mRNA in AN3-CA cells treated with miR-205-5p mimics. (**h**) Western blot analysis to detect E-cadherin protein in AN3-CA cells treated with miR-205-5p mimics. The data shown represent mean ± SEM from three independent experiments (*n* = 3). * *p* < 0.05, ** *p* < 0.01.

**Figure 4 ijms-24-15149-f004:**
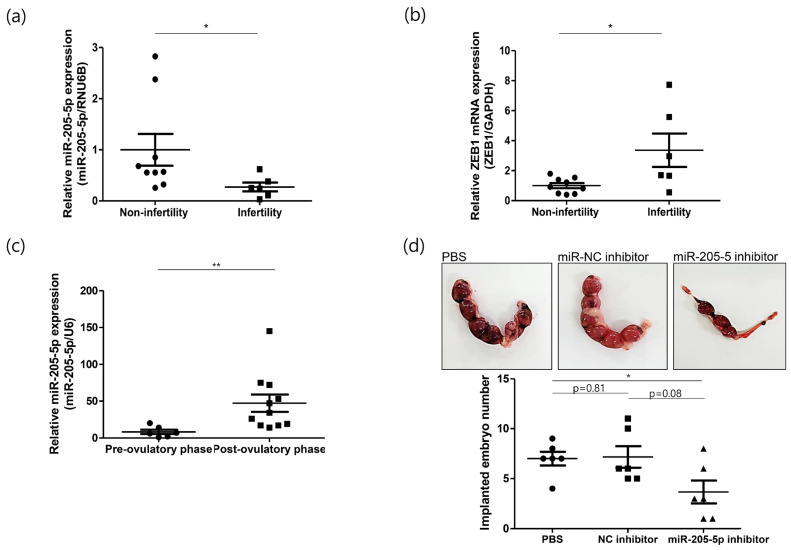
Role of miR-205-5p in embryo implantation. (**a**) Expression analysis of miR-205-5p in the endometrial samples of 9 non-infertile and 6 infertile women. (**b**) Quantitative RT-PCR analysis to detect ZEB1 mRNA in the endometrial samples of 9 non-infertile and 6 infertile women. (**c**) Expression analysis of miR-205-5p in the 6 pre-ovulatory and 11 post-ovulatory phase mouse uteri. (**d**) Analysis of implanted embryo area and number at 12.5 dpc after the injection of PBS, miR-NC inhibitor, or miR-205-5p inhibitor into a 2.5 dpc mouse uterus. Six mice were used per group. The data shown represent mean ± SEM. * *p* < 0.05, ** *p* < 0.01.

**Figure 5 ijms-24-15149-f005:**
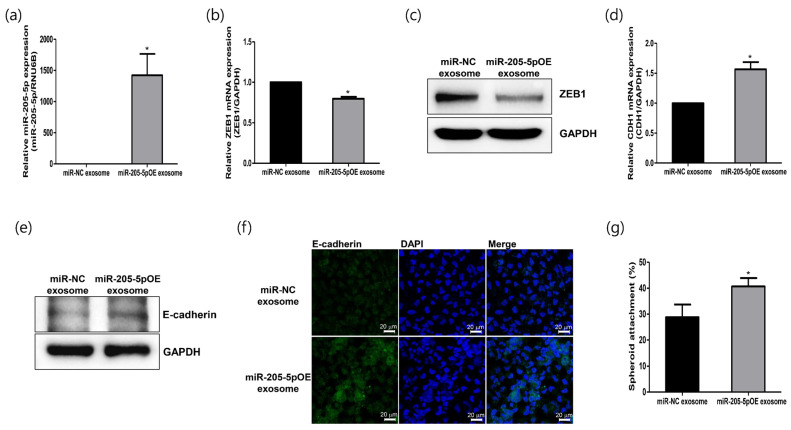
Exosomes overexpressing miR-205-5p mimics positively regulate endometrial receptivity in non-receptive endometrial epithelial cells. (**a**) Expression analysis of miR-205-5p in AN3-CA cells after treatment with exosomes harboring miR-205-5p mimics. (**b**) Analysis of ZEB1 mRNA using quantitative RT-PCR in AN3-CA cells after treatment with exosomes harboring miR-205-5p mimics. (**c**) Analysis of ZEB1 protein using Western blotting in AN3-CA cells after treatment with exosomes harboring miR-205-5p mimics. (**d**) Analysis of CDH1 mRNA using quantitative RT-PCR in AN3-CA cells after treatment with exosomes harboring miR-205-5p mimics. (**e**) Analysis of E-cadherin protein using Western blot analysis in AN3-CA cells after treatment with exosomes harboring miR-205-5p mimics. (**f**) Immunofluorescence image confirming E-cadherin expression in AN3-CA cells after treatment with exosomes harboring miR-205-5p mimics. Scale bars indicate 20 μm. (**g**) Spheroid attachment rate in AN3-CA cells treated with exosomes harboring miR-205-5p mimics. The data shown represent mean ± SEM from three independent experiments (*n* = 3). * *p* < 0.05.

**Figure 6 ijms-24-15149-f006:**
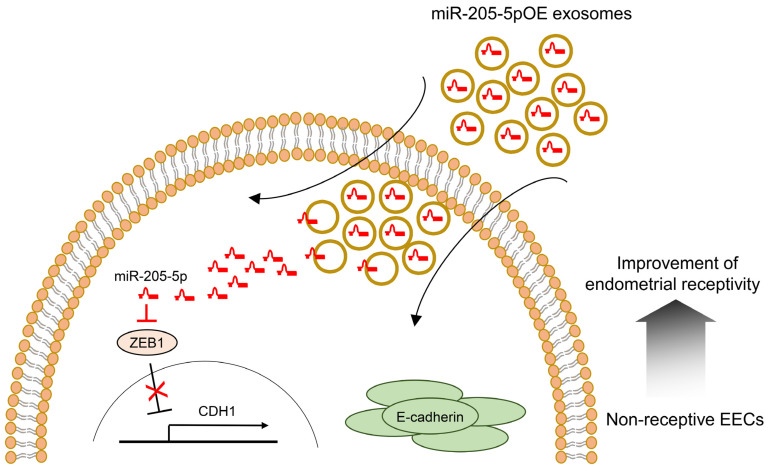
Proposed mechanism for improving endometrial receptivity by miR-205-5p-mediated ZEB1/E-cadherin pathway regulation. EEC, endometrial epithelial cell; OE, overexpressing.

**Table 1 ijms-24-15149-t001:** Top 20 differentially upregulated and downregulated miRNAs expressed in RL95-2 cells compared with those in AN3-CA cells.

	miRNA ID	Fold Change	*p*
Upregulated	hsa-miR-205-5p	14.02	1.77 × 10^−3^
	hsa-miR-205	12.61	1.51 × 10^−3^
	hsa-miR-934	9.78	6.91 × 10^−5^
	hsa-miR-203a-3p	9.21	2.14 × 10^−4^
	hsa-miR-141-3p	9.21	2.18 × 10^−3^
	hsa-miR-31-5p	9.04	1.94 × 10^−3^
	hsa-miR-200c-3p	8.97	3.33 × 10^−3^
	hsa-miR-200c	8.73	3.35 × 10^−3^
	hsa-miR-141	8.47	2.02 × 10^−3^
	hsa-miR-516-5p	8.44	2.73 × 10^−3^
	hsa-miR-203a	8.37	2.38 × 10^−4^
	hsa-miR-34a-5p	8.14	4.69 × 10^−3^
	hsa-miR-519a-5p	8.07	1.81 × 10^−4^
	hsa-miR-516a-1	8.06	2.32 × 10^−3^
	hsa-miR-516a-2	8.06	2.32 × 10^−3^
	hsa-miR-934	8.01	7.33 × 10^−5^
	hsa-miR-1283	7.83	1.98 × 10^−3^
	hsa-miR-31	7.65	8.52 × 10^−4^
	hsa-miR-34a	7.56	2.58 × 10^−3^
	hsa-miR-519a-3p	7.52	1.60 × 10^−4^
Downregulated	hsa-miR3180-5	−6.38	1.02 × 10^−3^
	hsa-miR-199b	−6.50	4.71 × 10^−2^
	hsa-miR-873-3p	−6.56	7.86 × 10^−3^
	hsa-miR-615	−6.70	2.89 × 10^−2^
	hsa-miR-873	−6.97	5.75 × 10^−3^
	hsa-miR-3180-3	−7.01	8.69 × 10^−4^
	hsa-miR-3180-1	−7.01	8.69 × 10^−4^
	hsa-miR-3180-2	−7.09	8.56 × 10^−4^
	hsa-miR-335	−7.10	7.12 × 10^−3^
	hsa-miR-335-5p	−7.12	7.80 × 10^−3^
	hsa-miR-335-3p	−7.26	1.90 × 10^−2^
	hsa-miR-296-3p	−7.80	3.42 × 10^−3^
	hsa-miR-615-3p	−7.98	2.11 × 10^−2^
	hsa-miR-125b-1-3p	−8.06	5.32 × 10^−3^
	hsa-miR-3180-3p	−8.45	6.71 × 10^−4^
	hsa-miR-873-5p	−8.49	3.82 × 10^−3^
	hsa-miR-3180	−8.57	7.17 × 10^−4^
	hsa-miR-196a-2	−8.96	6.22 × 10^−3^
	hsa-miR-196a-1	−9.14	5.60 × 10^−3^
	hsa-miR-196a-5p	−9.33	5.93 × 10^−3^

**Table 2 ijms-24-15149-t002:** Characteristics of the endometria of donors for quantitative reverse transcription polymerase chain reaction.

Variable/Group	Non-Infertility (20–24 mcd; *n* = 9)	Infertility (20–22 mcd; *n* = 6)	*p*
Age (y)	37.8 ± 2.6	38.5 ± 3.7	0.76
BMI (kg/m^2^)	22.7 ± 2.8	23.0 ± 4.4	0.95
No. of live births (*n*)	1.9 ± 0.4	0.2 ± 0.4	0.0015
No. of abortions (*n*)	0.4 ± 0.5	0.8 ± 1.3	0.82

*p*-values were determined using a *t*-test. Data are presented as mean ± SD.

**Table 3 ijms-24-15149-t003:** Primer sequences used for quantitative reverse transcription polymer chain reaction.

Gene	Primer Sequence	Annealing Temperature	Size of Amplicon (bp)
*ZEB1*	Forward: 5′-AAGAATTCACAGTGGAGAGAAGCC-3′ Reverse: 5′-CGTTTCTTGCAGTTTGGGCAT-3′	56 °C	51
*CDH1*	Forward: 5′-TCAGCGTGTGTGACTGTGAA-3′ Reverse: 5′-CCTCCAAGAATCCCCAGAAT-3′	56 °C	100
*GAPDH*	Forward: 5′-ACAGTCAGCCGCATCTTCTT-3′ Reverse: 5′-ACGACCAAATCCGTTGACTC-3′	56 °C	94

## Data Availability

Not applicable.

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
