# Peer review of "Exosomal miR-205-5p Improves Endometrial Receptivity by Upregulating E-Cadherin Expression through ZEB1 Inhibition"

_ijms, 2023, doi:10.3390/ijms242015149_

Round 1

Reviewer 1 Report

In this manuscript, the authors found that miR-205-5p was the most highly expressed miRNA in exosomes from receptive RL95-2 cells and it related to endometrial receptivity. In vivo experiments in mice showed that miR-205-5p expression was enhanced in the post-ovulatory phase and its inhibitor reduced embryo implantation. In addition, authors also show that modified exosomes overexpressing miR-205-5p mimic increased E-cadherin expression by targeting ZEB1 and clearly improved spheroid attachment of non-receptive AN3-CA cells. This paper is organized well. I have several specific concerns regarding this paper, as outlined below.

Comments

1.       Fig2 c and d, both are relatively miRNA expression level. The scale of Y axis is not understood. Is it a different analysis method?

2.       Fig3 b, what is the location of this target fragment? Figure 3c,d, clarify the “relative expression ”.

3.       Fig 4c, clarify the “relative expression. Also, Fig5a”

4.       Fig 4, how many subjects are in each group? n=?

5.       Fig5 how long does the treatment take?

Author Response

Reviewer #1: In this manuscript, the authors found that miR-205-5p was the most highly expressed miRNA in exosomes from receptive RL95-2 cells and it related to endometrial receptivity. In vivo experiments in mice showed that miR-205-5p expression was enhanced in the post-ovulatory phase and its inhibitor reduced embryo implantation. In addition, authors also show that modified exosomes overexpressing miR-205-5p mimic increased E-cadherin expression by targeting ZEB1 and clearly improved spheroid attachment of non-receptive AN3-CA cells. This paper is organized well. I have several specific concerns regarding this paper, as outlined below.

  1. Fig2 c and d, both are relatively miRNA expression level. The scale of Y axis is not understood. Is it a different analysis method?

Response: We thank you for the comment. Analysis of miRNA expression was performed using the TaqMan miRNA method as described in section 4.7. in the Materials and Methods. The scale of Y axis in Figure 2c and d indicates fold difference in miRNA levels in receptive RL95-2 cells or the derived exosomes compared with the levels in the non-receptive AN3-CA control group. Specially, miR-205-5p was dominantly expressed at 6 × 105 fold in RL95-2 cells. Additionally, Figure 2c displays the upregulated miRNAs in receptive RL95-2 cells or the derived exosomes compared with those in the non-receptive control group, and Figure 2d shows the downregulated miRNAs in RL95-2 cells or the secreted exosomes.

  1. Fig3 b, what is the location of this target fragment? Figure 3c,d, clarify the “relative expression ”.

Response: We thank you for the comment. The red sequence in Figure 3b represents the binding site of miR-205-5p between nt 4470 and 4477 in the 3'UTR of ZEB1 (NM_001128128). We have indicated the binding site in the legend of Figure 3. Figure 3c indicates the normalized luciferase activity of ZEB1-3'UTR-WT and -MUT compared with that of the control (psiCHEK2.0). The expression levels of the four selected miRNAs were normalized using a reference miRNA (RNU6B). The expression of each target miRNA was quantified relative to reference miRNA expression in AN3-CA cells or exosomes derived from AN3-CA cells. Figure 3d shows the level of miR-205-5p expression after normalization to the reference miRNA (RNU6B) in cells overexpressing miR-205-5p.

  1. Fig 4c, clarify the “relative expression. Also, Fig5a”

Response: We thank you for the suggestion. As mentioned above, Figure 4c displays the normalized miR-205-5p expression compared with the reference miRNA (U6) known as reference miRNA of mouse. Figure 5a indicates the relative expression level of miR-205-5p compared to the reference miRNA level (RNU6B).

  1. Fig 4, how many subjects are in each group? n=?

Response: We thank you for the comment. The number of samples used in the analysis presented in Figure 4 is indicated with dots. For the analysis presented in Figure 4a and b, we used 9 samples from non-infertile patients and 6 from infertile patients. We also used 6 pre-ovulatory phase and 11 post-ovulatory phase samples for the analysis presented in Figure 4c. For the analysis shown in Figure 4d, six mice were used in each group. The number of samples used in the experiments has been indicated in the legend of Figure 4.

  1. Fig5 how long does the treatment take?

Response: We thank you for this comment. The cells were treated with exosomes for 48 h and co-cultured JAr spheroid for 6 h. Th duration of treatment has been indicated in section 4.8 in the Materials and methods (page 14, line 472).  

Reviewer 2 Report

In this study, Yu et al. first profiled the exosomes isolated from a receptive and a non-receptive endometrial carcinoma cell lines. They then selected miR-205-5p as a target for this study. Subsequently by using in-silico method, they choose ZEB1 as the target for miR-205-5p, and further claimed that CDH1 is the effector regulating the adhesiveness of JAr spheroids. They also analyzed human endometrial samples to correlate the differential expression patterns of these molecules in “fertile” and “non-fertile” women. They further used mouse model and genetically modified exosomes to confirm the role of miR-205-5p/ZEB1/CDH1 during implantation. Although the study provides some novel findings, the evidence of the proposed miR-205-5p/ZEB1/CDH1 axis during implantation needs further clarification and data support before it can be published. Major issues are listed below,

Major issues:

1.     The basic question of why the endometrial epithelial cells need to export miR-205-5p via exosomes and regulate the expression of ZEB1 of the endometrial cell itself should be addressed. Any postulation that why the whole process cannot be carried out within the cells.

2.     The detailed information of human endometrial samples is missing, including details of demographic data. Are these IVF patients? How to define fertile and infertile? Reasons of infertility? The details of how to recruit the fertile patients should also be included.

3.     Line 227 “expression pattern in endometrial samples”, did the authors used the whole tissue to do the analysis? How can the expression in the whole tissue which includes several cell types represent the expression in the exosomes?

4.     As the current study highlighted the importance of exosome secreted from endometrium to facilitate the embryo attachment, comparison of miR-205-5p should be performed on the uterine fluid from the infertile and fertile patients. Has miR-205-5p been reported to be expressed in the published dataset of human uterine fluid?  

5.     The authors repeatedly emphasized the miR-205-5p/ZEB1/CDH1 axis in all the experiments. However, the direct link is still missing. miR-205-5p can act on hundreds of targets in the endometrial cells. A proper experiment with ZEB1 knockdown on CDH1 expression and spheroid attachment should be included in this study. To further confirm the miR-205-5p/ZEB1/CDH1 axis, a rescue experiment with the transfection of miR-205-5p and ZEB1 overexpression on CDH1 expression and spheroid attachment should also be included. Otherwise, the author can only claim for correlation of the axis.  

6.     Line 104-105: the expression level of CD63, but not CD9, differed between the two cell lines. Why? How can it be concluded that “cell-derived exosomes of high quality and purity were successfully isolated.”?

7.     Fig1d: The details of exosome treatment on trophoblast spheroid attachment is missing. How long had the cells been treated? Any control? AN3-CA isolated exosomes should be the proper control for this experiment but missing. If the biological difference between AN3-CA and RL-95 derived EV cannot be confirmed at the beginning, how can the authors proceed to do miRNA sequencing of the two cell types?

            8) “adheren junction” seem not the one with highest pathway enrichment for               miR-205-5p only, but miR-200c-3p too. Explanation on the selection should               be included. 

Average

Author Response

Reviewer #2: In this study, Yu et al. first profiled the exosomes isolated from a receptive and a non-receptive endometrial carcinoma cell lines. They then selected miR-205-5p as a target for this study. Subsequently by using in-silico method, they choose ZEB1 as the target for miR-205-5p, and further claimed that CDH1 is the effector regulating the adhesiveness of JAr spheroids. They also analyzed human endometrial samples to correlate the differential expression patterns of these molecules in “fertile” and “non-fertile” women. They further used mouse model and genetically modified exosomes to confirm the role of miR-205-5p/ZEB1/CDH1 during implantation. Although the study provides some novel findings, the evidence of the proposed miR-205-5p/ZEB1/CDH1 axis during implantation needs further clarification and data support before it can be published. Major issues are listed below, 

MAJOR:

  1. The basic question of why the endometrial epithelial cells need to export miR-205-5p via exosomes and regulate the expression of ZEB1 of the endometrial cell itself should be addressed. Any postulation that why the whole process cannot be carried out within the cells.

Response: Thank you for the valuable comment. Exosomes secreted from cells are capable of delivering cargos such as proteins, lipids, and nucleic acids to the recipient cells distal from their release site and represent a novel mode of intercellular communication such as immune response, signal transduction, and antigen presentation. Therefore, exosomes have been attracting attention as a clinical therapeutic through material transfer (Zhang et al., 2019). In this study, we sought to determine the possibility of improving endometrial receptivity of non-receptive endometrial epithelial cells (EECs) using exosomes containing high levels of miR-205-5p derived from receptive EECs. As mentioned in Discussion (page 12, lines 356-361), our focus was not on the effect of miR-205 within EECs-derived exosomes on their own cells, but rather on exploring the clinical application of exosome-based therapy in the context of implantation failure.

  1. The detailed information of human endometrial samples is missing, including details of demographic data. Are these IVF patients? How to define fertile and infertile? Reasons of infertility? The details of how to recruit the fertile patients should also be included.

Response: Thank you for this important point. Characteristics of the donor endometrium are shown in Table 2. Infertility is defined as the inability of a couple to conceive naturally even after 1 year of regular unprotected sexual intercourse (Kamel et al., 2010). Infertile patients were selected according to this definition, and their diagnosis history was as follows. Two of six selected infertile patients experienced two or more repeated miscarriages. Three patients had at least three high-quality embryos implanted into the uterus but did not become pregnant. Therefore, they were diagnosed with recurrent implantation failure. In addition, one patient was diagnosed with recurrent miscarriages and recurrent implantation failures. Non-infertile patients were recruited for a clinical trial at Konyang University Hospital, and they provided written informed consent for the collection of endometrial tissues. These details are presented in section 4.1 in the Materials and Methods (page 12, lines 376-384).

  1. Line 227 “expression pattern in endometrial samples”, did the authors used the whole tissue to do the analysis? How can the expression in the whole tissue which includes several cell types represent the expression in the exosomes?

Response: We thank you for this comment. Endometrial samples contain multiple cell types as you mentioned. It is almost impossible to isolate only endometrial epithelial cells from a small amount of endometrium. Therefore, we isolated total RNA in endometrium samples to identify miR-205-5p expression in the endometria. The increased miR-205-5p level in non-infertile women is consistent with its upregulated expression in receptive cells rather than non-receptive endometrial epithelial cells. Therefore, regulation of endometrial receptivity may be associated with changes in the expression of miR-205-5p.

  1. As the current study highlighted the importance of exosome secreted from endometrium to facilitate the embryo attachment, comparison of miR-205-5p should be performed on the uterine fluid from the infertile and fertile patients. Has miR-205-5p been reported to be expressed in the published dataset of human uterine fluid?

Response: We agree with the suggestion. We did not obtain uterus fluids when endometrial samples were collected. However, Yang et al. (2023:37240034) recently reported the expression of miR-205-5p in endometrial fluid. Therefore, we inserted the following sentence in the Discussion: “Recently, Yang et al. (2023) reported miR-205-5p expression in uterus fluid” (page 10, lines 290-291).  

  1. The authors repeatedly emphasized the miR-205-5p/ZEB1/CDH1 axis in all the experiments. However, the direct link is still missing. miR-205-5p can act on hundreds of targets in the endometrial cells. A proper experiment with ZEB1 knockdown on CDH1 expression and spheroid attachment should be included in this study. To further confirm the miR-205-5p/ZEB1/CDH1 axis, a rescue experiment with the transfection of miR-205-5p and ZEB1 overexpression on CDH1 expression and spheroid attachment should also be included. Otherwise, the author can only claim for correlation of the axis.

Response: We thank you for the valuable comment on our manuscript. Unfortunately, ZEB1 siRNA and ZEB1 expression vector are not available. It requires a lot of time to perform the experiments suggested by you. However, there is a report on the relationship between ZEB1 and E-cadherin in endometrial epithelial cells. Xiao et al. (2019) reported that the inhibition of ZEB1 expression induces E-cadherin expression in endometrial epithelial cells such as Ishikawa and HEC-1B cells, as shown below. Consistent with our results, Xiao et al. (2019) suggest a negative relationship between ZEB1 and CDH1 in endometrial epithelial cells. Therefore, the function of ZEB1 in regulating E-cadherin expression in endometrial epithelial cells was confirmed.
